# Normal mitochondrial function in *Saccharomyces cerevisiae* has become dependent on inefficient splicing

Marina Rudan[1], Peter Bou Dib[2], Marina Musa[1], Matea Kanunnikau[1], Sandra Sobočanec[3], David Rueda[4,5,6], Tobias Warnecke[4,5]*, Anita Kriško[1]*

[1]Mediterranean Institute for Life Sciences, Split, Croatia; [2]Institut für Zellbiochemie, Universitätsmedizin Göttingen, Göttingen, Germany; [3]Division of Molecular Medicine, Rudjer Boškovic Institute, Bijenička, Zagreb, Croatia; [4]MRC London Institute of Medical Sciences, London, United Kingdom; [5]Institute of Clinical Sciences, Faculty of Medicine, Imperial College London, London, United Kingdom; [6]Molecular Virology, Faculty of Medicine, Imperial College London, London, United Kingdom

**Abstract** Self-splicing introns are mobile elements that have invaded a number of highly conserved genes in prokaryotic and organellar genomes. Here, we show that deletion of these selfish elements from the *Saccharomyces cerevisiae* mitochondrial genome is stressful to the host. A strain without mitochondrial introns displays hallmarks of the retrograde response, with altered mitochondrial morphology, gene expression and metabolism impacting growth and lifespan. Deletion of the complete suite of mitochondrial introns is phenocopied by overexpression of the splicing factor Mss116. We show that, in both cases, abnormally efficient transcript maturation results in excess levels of mature *cob* and *cox1* host mRNA. Thus, inefficient splicing has become an integral part of normal mitochondrial gene expression. We propose that the persistence of *S. cerevisiae* self-splicing introns has been facilitated by an evolutionary lock-in event, where the host genome adapted to primordial invasion in a way that incidentally rendered subsequent intron loss deleterious.
DOI: https://doi.org/10.7554/eLife.35330.001

**\*For correspondence:**
tobias.warnecke@lms.mrc.ac.uk (TW);
anita.krisko@medils.hr (AK)

**Competing interests:** The authors declare that no competing interests exist.

## Introduction

Mobile genetic elements frequently compromise host fitness (*Werren, 2011*), corrupting genetic information or disturbing adaptive gene expression patterns, sometimes to lethal effect. Despite this, mobile genetic elements are ubiquitous in most eukaryotic genomes (*Hurst and Werren, 2001*). How do these selfish elements persist in a genomic environment where – even in the absence of selection – mutational forces constantly work to erode them? Although mobile elements can donate motifs (or domains) that are co-opted into host regulatory pathways (or genic sequence) over time (*Feschotte, 2008*), deletions usually whittle away all but the core beneficial motif. The components that once mediated mobility, such as the reverse transcriptases of long interspersed nuclear elements (LINEs), are typically lost. Thus, functional selfish elements that remain mobile are thought to persist over evolutionary time not by virtue of sporadic beneficial effects for the host, but because they replicate and spread to other sites in the genome faster than they are deleted (*Feschotte, 2008*). The element survives, not where it originally invaded but as a descendant copy elsewhere in the genome.

An interesting exception in this regard are self-splicing introns, which populate some highly expressed genes in archaea, bacteria, and organellar genomes of fungi and plants (*Lambowitz and*

*Belfort, 1993*). In contrast to other mobile elements, self-splicing introns do not spawn a large pool of copies that disperse across the genome to escape mutational erasure. Rather, owing to highly specific homing sites, each intron is typically confined to a single location in the host genome and evolutionary persistence seems to rely on continued re-invasion, either from other individuals in the same population (*Goddard and Burt, 1999*) or across species boundaries (*Skelly and Maleszka, 1991*; *Dujon, 1989*; *Repar and Warnecke, 2017*). Self-splicing introns can spread despite considerable fitness costs to the host (*Hickey, 1982*). However, in practice, fitness costs might be relatively low as the host RNA is intact and fully functional once the intron has been spliced out. As far as we know, self-splicing introns do not contribute positively to host fitness and, naively, one would expect that deleting these introns from the genome would be beneficial or, at worst, make no difference to the host.

Here, we investigate the consequences of deleting all 13 self-splicing introns from the *S. cerevisiae* mitochondrial genome, where they reside in three host genes: the 21S ribosomal RNA gene Q0158 (which harbours a single group I intron named *omega*) and two protein-coding genes, *cox1* (group I: aI3, aI4, aI5α, aI5β; group II: aI1, aI2, aI5γ) and *cob* (group I: bI2, bI3, bI4, bI5; group II: bI1), both encoding components of the electron transport chain. Note here, that we use 'self-splicing introns' as a convenient shorthand to describe the complete collection of group I and group II introns, even though timely splicing in vivo often depends on one or several *trans*-factors (see below) (*Lambowitz and Belfort, 1993*). We show that, contrary to expectations, removing these introns has dramatic consequences for mitochondrial physiology and function, triggering changes in nuclear gene expression that affect organismal growth and lifespan. Our results demonstrate that the presence of mitochondrial self-splicing introns has become integral to normal mitochondrial gene expression and that, curiously, normal mitochondrial function in *S. cerevisiae* has come to require inefficient splicing. Our findings have implications for understanding how self-splicing introns and mobile elements more generally can survive over evolutionary time without providing an adaptive benefit to the host.

## Results

### Removal of mitochondrial introns is associated with a multi-faceted stress phenotype

To test whether the removal of self-splicing mitochondrial introns affects host physiology and fitness, we compared two *S. cerevisiae* strains that are isogenic with regard to their nuclear genomes (except for a single marker gene, *ura3*, see Materials and methods) but differ with respect to mitochondrial intron content. The control strain a161 (WT) contains the full complement of seven *cox1* and five *cob* introns, whereas strain a161-U7 ($I_0$) carries an intronless mitochondrial genome, which was originally constructed by Seraphin and co-workers via serial recombination of natural yeast isolates that lack individual mitochondrial introns (*Séraphin et al., 1987*).

Contrary to a model where self-splicing introns are dispensable parasitic passengers, we find that $I_0$ exhibits stark phenotypic differences to the control strain. When the two strains are cultured in isolation, exponential growth on glucose-supplemented YPD medium is ~30% slower for $I_0$ compared to WT (*Figure 1a*). $I_0$ also fares poorly when pitted directly against WT in competitive fitness assay (*Figure 1—figure supplement 1*). Chronological life span (CLS, see Materials and methods), on the other hand, is almost two-fold longer for $I_0$ (*Figure 1b*). At the cellular level, $I_0$ displays increased mitochondrial mass and volume and a mitochondrial morphology characterized by a large network of branched tubules of homogeneous diameter (*Figure 1c–e*). Transcript levels of mitofusin (*fzo1*) and the GTPase *mgm1*, key nuclearly encoded regulators of mitochondrial fusion, are strongly upregulated (qPCR, *t*-test, *fzo1*: 4.02-fold, p=0.002; *mgm1*: 5.24-fold, p=3.19×10⁻⁵, *Figure 2a*) while levels of *dnm1* and *fis1*, which orchestrate mitochondrial fission, are only moderately induced (qPCR, *t*-test, *dnm1*: 1.67-fold, p=0.026; *dnm1*: 1.81-fold, p=0.085), suggesting that the changes in mitochondrial morphology result from increased fusion rather than impaired fission. The $I_0$ strain also exhibits a 2.7-fold increase in mitochondrial DNA copy number (qPCR, *t*-test p=1.09×10⁻⁷). At the same time, there are no significant differences in mitochondrial inner membrane potential, as measured by 3,3'-Dihexyloxacarbocyanine iodide [DiOC6(3)] fluorescence (*Figure 1f*), suggesting that mitochondria are functional despite grossly altered morphology. This notion is further supported by

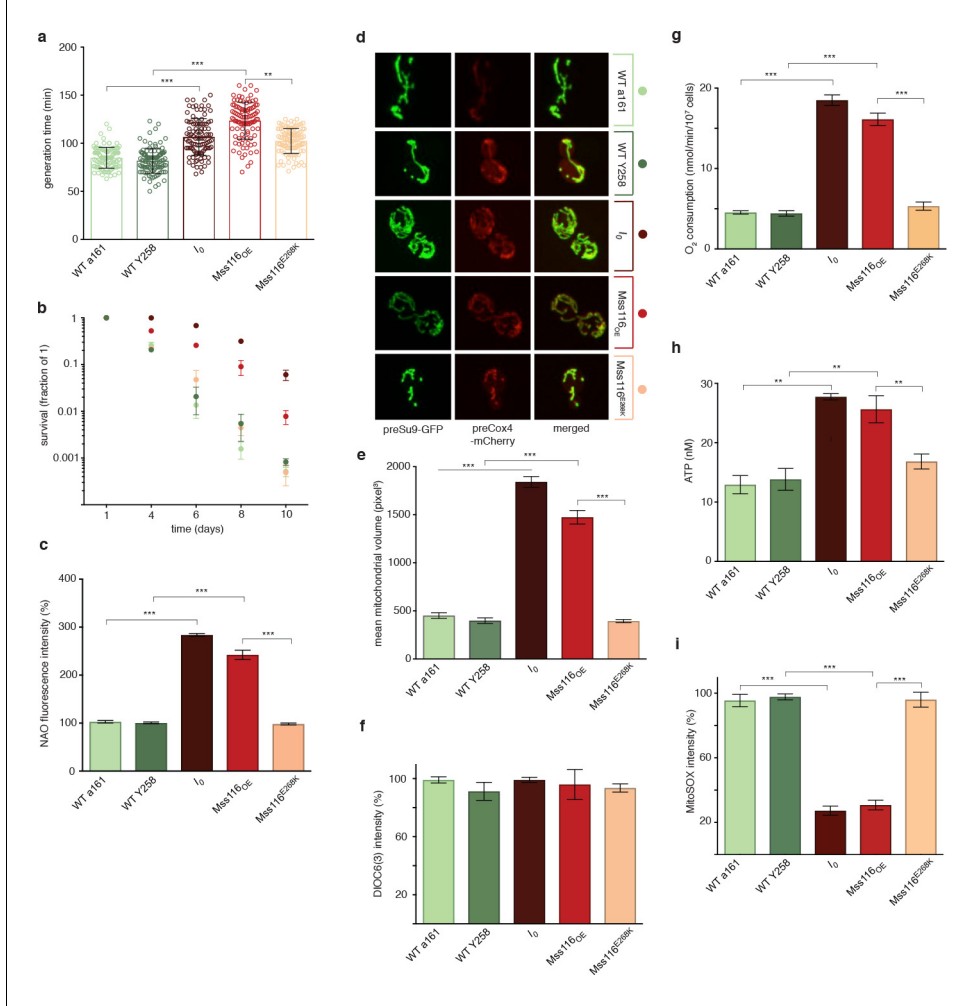

**Figure 1.** Phenotypic effects of deleting all self-splicing introns from the *S. cerevisiae* mitochondrial genome. Deletion of mitochondrial introns ($I_0$) or overexpression of Mss116 ($Mss116_{OE}$) (a) reduces growth rates, (b) extends chronological lifespan, (c) increases mitochondrial mass, measured as NAO fluorescence, (d,e) increases mitochondrial volume, (g) oxygen consumption, and (h) ATP levels but (i) decreases superoxide levels, measured as MitoSOX fluorescence. (f) Mitochondrial inner membrane potential does not differ significantly between strains. WT a161 and WT Y258 are control strains for $I_0$ and $Mss116_{OE}$, respectively, as described in the text. The $Mss116^{E268K}$ strain harbours a mutant version of Mss116 that lacks ATPase activity. As a visual guide, strains are colored consistently throughout. Bar heights display the mean of three biological replicates, each calculated as the mean of three technical replicates. Error bars are standard errors of the mean. ***p<0.001; **p<0.01; *p<0.05 (ANOVA plus post hoc).

DOI: https://doi.org/10.7554/eLife.35330.002

The following figure supplements are available for figure 1:

**Figure supplement 1.** Competitive fitness is decreased in $I_0$ (competed against WT a161) and $Mss116_{OE}$ (competed against the empty vector control WT Y258).

DOI: https://doi.org/10.7554/eLife.35330.003

**Figure supplement 2.** $I_0$ and $Mss116_{OE}$ show no qualitative difference in growth on (a) glucose and (b) glycerol.

DOI: https://doi.org/10.7554/eLife.35330.004

**Figure supplement 3.** Mss116 expression level is 2.5-fold increased in the $Mss116_{OE}$ strain.

DOI: https://doi.org/10.7554/eLife.35330.005

**Figure supplement 4.** Median and maximum replicative lifespan of $Mss116_{OE}$ is extended compared to the empty vector control and $Mss116^{E268K}$.

DOI: https://doi.org/10.7554/eLife.35330.006

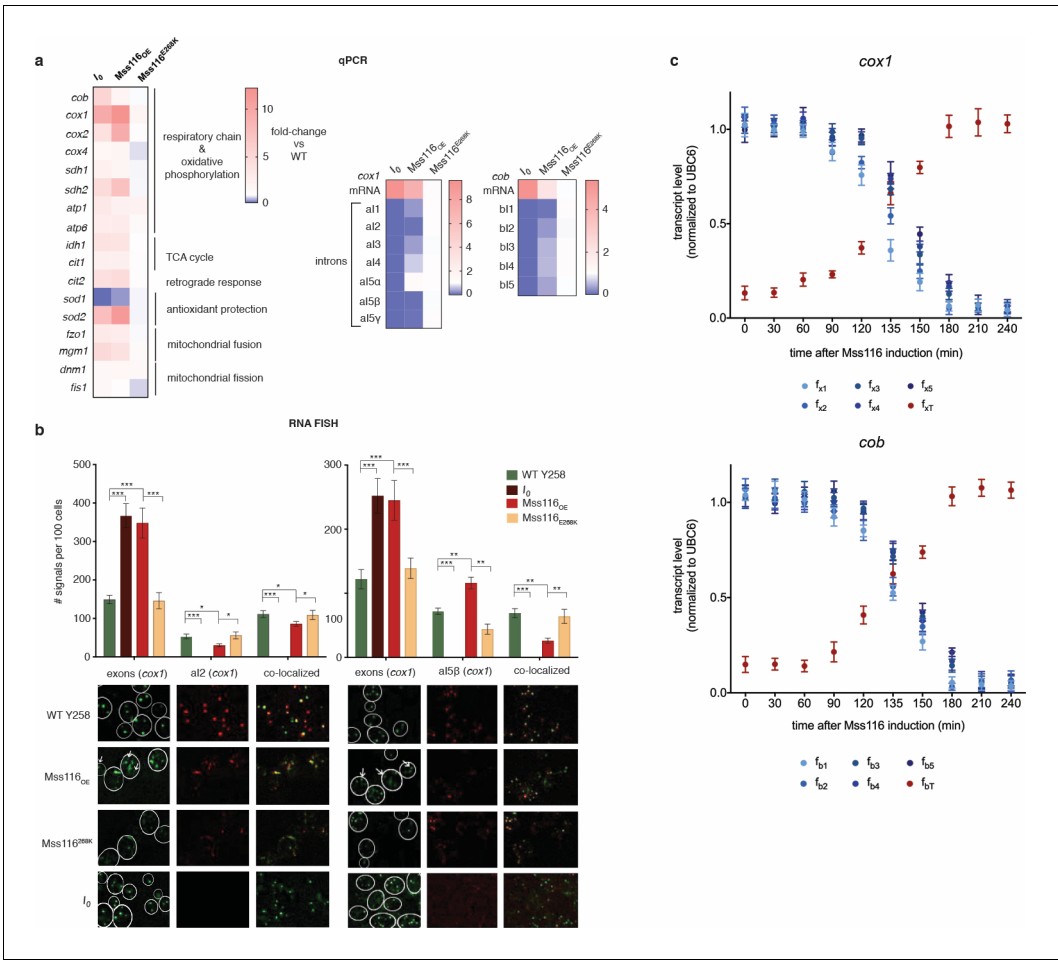

**Figure 2.** RNA abundance changes associated with intron removal. (a) qPCR measurements of selected genes, comparing focal strains ($I_0$, $Mss116_{OE}$, $Mss116^{E268K}$) to their isogenic control strains (left panel). For both *cox1* (central panel) and *cob* (right panel), intron levels are specifically reduced upon overexpression of $Mss116_{OE}$ but not $Mss116^{E268K}$, while mature mRNA levels increase. Heat maps display mean values of log-fold changes observed across three biological replicates (each averaged over three technical replicates). UBC6 was used for normalization. (b) RNA-FISH confirms elimination/reduction of introns aI2 and aI5β from the *cox1* transcript pool. Exon (green), intron (red) and co-localized (green/red) puncta were counted in more than 300 cells. The bar chart shows the number of signals per 100 cells. Bar heights display the mean of three biological replicates (each averaged over three technical replicates). Error bars are standard error of the mean. ***p<0.001; **p<0.01; *p<0.05 (ANOVA plus post hoc). White lines mark cell boundaries. White arrows mark examples of exonic puncta that do not co-localize with intronic puncta. (c) qPCR time series of pre-mRNA and mature mRNA levels following induction of Mss116. Mature mRNA for *cox1* and *cob* was quantified using primer pairs ($f_{xT}$, $f_{bT}$) overlapping the terminal exon-exon junctions. Pre-mRNA was quantified using a series of primer pairs ($f_{x1-5}$, $f_{b1-5}$). For each pair, one primer is located in exonic, the other in intronic sequence, as detailed in *Figure 2—figure supplement 4*. Each circle (shades of blue for the pre-mRNA and red for the mature transcript) represents the mean value from three biological replicates (each averaged over three technical replicates). UBC6 was used for normalization.
DOI: https://doi.org/10.7554/eLife.35330.007

The following figure supplements are available for figure 2:

**Figure supplement 1.** Mss116 overexpression does not affect mitochondrial function in the $I_0$ strain.
DOI: https://doi.org/10.7554/eLife.35330.008
**Figure supplement 2.** Mss116 deletion does not affect $I_0$ phenotypes, highlighted by (a) the transcript levels (measured by qPCR) of relevant genes, (b) mitochondrial morphology, and (c) mitochondrial volume.
DOI: https://doi.org/10.7554/eLife.35330.009
**Figure supplement 3.** Expression level of Mss116 following induction using 2% galactose.
DOI: https://doi.org/10.7554/eLife.35330.010

*Figure 2 continued on next page*

*Figure 2 continued*

**Figure supplement 4.** Schematic representation of the fragments of *cox1* and *cob* amplified to determine the pre-mRNA and mature mRNA levels.
DOI: https://doi.org/10.7554/eLife.35330.011

the observation that $I_0$ retains the capacity to grow on glycerol, a non-fermentable carbon source (*Figure 1—figure supplement 2*), as previously reported for a different nuclear background (*Minczuk et al., 2002*). In fact, biomarkers of mitochondrial metabolism point to increased mitochondrial activity, with higher oxygen consumption (*Figure 1g*) and cellular ATP levels (*Figure 1h*) during exponential growth. Despite increased activity, levels of mitochondrial superoxide are reduced (*Figure 1i*), likely reflecting a > 7.5-fold upregulation of the mitochondrial ROS-scavenger *sod2* (qPCR, *t*-test p=0.003, *Figure 2a*). Thus, intron removal challenges but does not terminally compromise mitochondrial function.

## Removal of mitochondrial introns triggers the retrograde response

The phenotypic changes we observe suggest an involvement of the retrograde response, as do upregulation of *cit2*, upregulation of the two rate-limiting members of the TCA cycle, *cit1* and *idh1*, and upregulation of both mitochondrially (*cox1*, *cox2*, *atp6*) and nuclearly (*cox4*, *atp1*, *sdh1*, *sdh2*) encoded parts of the respiratory chain (*Figure 2a*). Indeed, deletion of *rtg2*, the transcriptional master regulator of the retrograde response and a sensor of mitochondrial dysfunction, suppresses the $I_0$ phenotype (*Figure 3*). Tubular structure is lost and large spherical shapes become prominent (*Figure 3a*), suggesting distinct defects in the maintenance of mitochondrial ultrastructure (*Paumard et al., 2002*; *Velours et al., 2009*). Deletion of *rtg2* in WT strains, where the retrograde response is not activated, has no significant effect on mitochondrial volume and morphology, oxygen consumption and ATP levels (*Figure 3a–d*). The extension of CLS is abrogated in the absence of *rtg2*, becoming shorter even than the wildtype (*Figure 3e*). Upon deletion of *hap4*, the transcriptional activator of nuclearly encoded components of the respiratory chain (as well as the TCA cycle enzymes under normal conditions), mitochondrial morphology reverts back to the wild-type state (*Figure 4a–e*). We conclude that an intact retrograde response, including upregulation of nuclear components of the respiratory chain, is necessary to generate the mitochondrial phenotype observed in the $I_0$ strain.

## *Promoter attenuation of* cox1 *and* cob *reverses the phenotype*

To determine the ultimate molecular trigger(s) of the retrograde response, we examined how intron removal affects host gene expression. As previous work had found the *omega* intron to be 'optional' – present in some yeast strains but absent in others without obvious phenotypic effects (*Wolters et al., 2015*; *Dujon, 1980*) – we focused on *cox1* and *cob*. For both these genes, mRNA levels are strongly elevated in $I_0$ (10.9-fold and 5.8-fold for *cox1* and *cob*, respectively; *Figure 2a*) but also in Δ*rtg2* (*Figure 3f*) and Δ*hap4* (*Figure 4f*), suggesting that this is a direct effect of intron removal rather than a downstream consequence of activating the retrograde response. To investigate whether elevated *cox1* and/or *cob* transcript levels might underpin the wider transcriptional, metabolic and phenotypic changes, we introduced attenuating point mutations into the promoters of *cox1* and *cob* (see Materials and methods). We find that simultaneous attenuation of both promoters in $I_0$ fully reverses the $I_0$-characteristic suit of morphological and molecular phenotypes (see $I_0pp$ throughout *Figure 5*). Attenuation of either *cob* ($I_0cobp$) or *cox1* ($I_0cox1p$) in isolation only partially reverses the phenotype, although $I_0cobp$ has a larger relative effect than $I_0cox1p$ (*Figure 5*).

## The effects of intron removal are phenocopied by overexpression of Mss116

Next, we sought to establish why the wholesale deletion of self-splicing introns leads to increased abundance of the host transcripts. We considered two main possibilities. First, in deleting introns, we might have inadvertently removed DNA-/RNA-level regulatory elements that affect expression of the host genes. Alternatively, the act of short-circuiting the splicing process itself might interfere with normal expression. Specifically, we hypothesized that normal levels of transcription might be

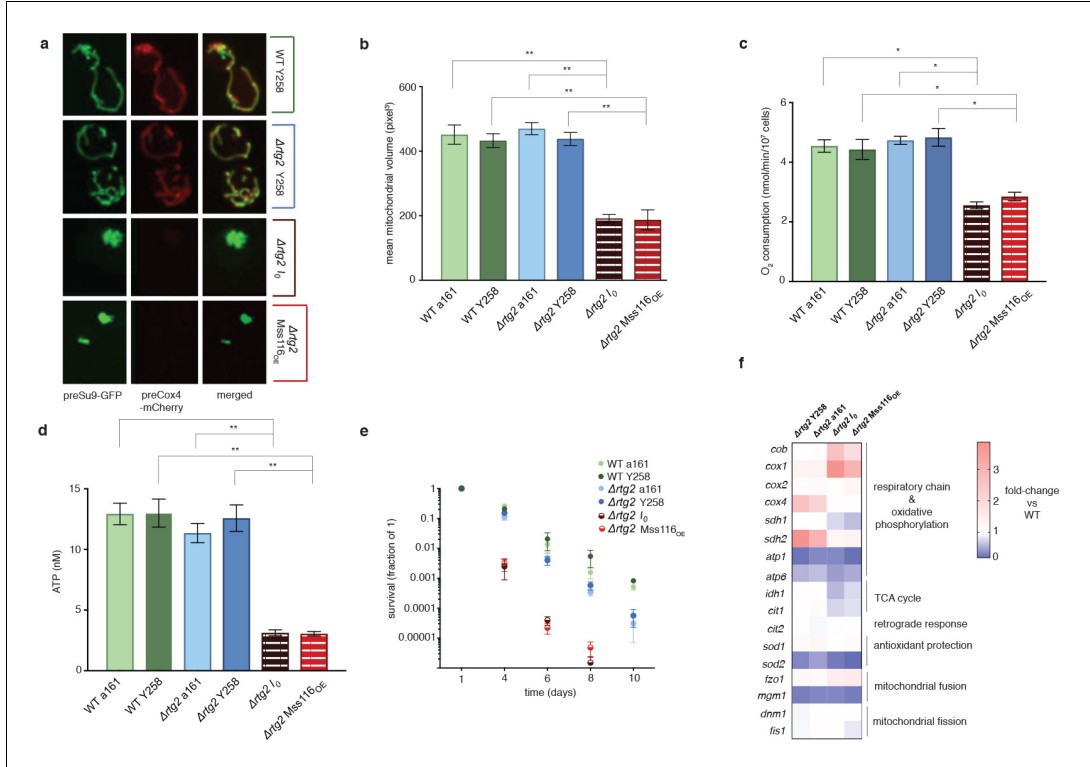

**Figure 3.** The intronless phenotype requires a functional retrograde response. (**a**) Mitochondrial morphology is altered and (**b**) mitochondrial volume, (**c**) oxygen consumption, (**d**) ATP levels, and (**e**) chronological lifespan are reduced when $rtg2$ is deleted in the $I_0$ or $Mss116_{OE}$ background. This contrasts sharply with $I_0$ and $Mss116_{OE}$ where $rtg2$ is intact (see **Figure 1**). Bar heights display the mean of three biological replicates (each averaged over three technical replicates). Error bars are standard error of the mean. ***$p<0.001$; **$p<0.01$; *$p<0.05$ (ANOVA plus post hoc). (**f**) Transcriptional responses in different strains where $rtg2$ has been deleted, as measured by qPCR. Heat maps display mean values of log-fold changes observed across three biological replicates (each averaged over three technical replicates). UBC6 was used for normalization.

DOI: https://doi.org/10.7554/eLife.35330.012

tuned to accommodate a certain proportion of transcripts that fail to splice correctly. Group II introns in particular are known for low splicing efficiency, even in the presence of auxiliary proteins (**Karunatilaka et al., 2010**). As a corollary, a large fraction of pre-mRNAs might be targeted and degraded by mitochondrial quality control, either because splicing is erroneous (mis-splicing) or does not occur in a timely manner so that the transcript is shunted into degradation (kinetic coupling). In $I_0$, splicing does not occur so that erroneous splicing products do not arise. As a result, production of functional $cox1/cob$ mRNAs might overshoot its target and trigger a system-wide response, for example because altered COX1/COB levels upset dosage balance amongst respiratory complexes. To test this abnormally-efficient-maturation hypothesis and simultaneously rule out that mitochondrial stress is caused by the removal of DNA-/RNA-level regulatory elements, we sought to alter splicing efficiency by orthogonal means. To this end, we overexpressed the nuclearly encoded DEAD box RNA helicase Mss116, which promotes splicing of all *S. cerevisiae* mitochondrial introns by remodeling or stabilizing splice-relevant RNA structures in an ATP-dependent manner (**Karunatilaka et al., 2010**; **Huang et al., 2005**). We confirmed overexpression and mitochondrial localization of Mss116 by flow cytometry and immunofluorescence, respectively, using an N-terminal His-tagged version of the protein (**Figure 1—figure supplement 3**), and then characterized the effects of Mss116 overexpression in the Y258 strain using an untagged version of the protein. Remarkably, the Mss116 overexpression strain ($Mss116_{OE}$) phenocopies $I_0$.

$Mss116_{OE}$ exhibits increased generation times, extended chronological life span, lower competitive fitness, increased mitochondrial fusion, 2.9-fold increased mtDNA copy number (qPCR, *t*-test

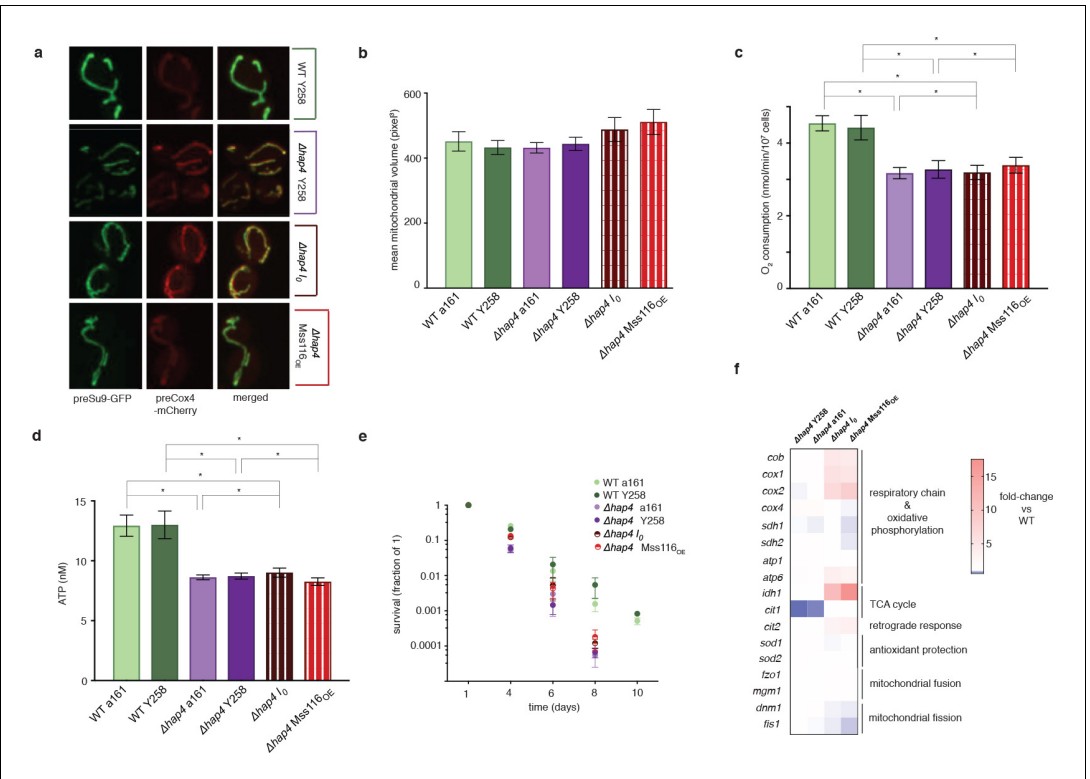

**Figure 4.** Hap4 is required for the intronless phenotype. (**a**) Mitochondrial morphology, (**b**) mitochondrial volume, (**c**) oxygen consumption, (**d**) ATP levels, and (**e**) chronological lifespan do not differ between $I_0$/$Mss116_{OE}$ and their corresponding control strains if *hap4* has been deleted. Bar heights display the mean of three biological replicates (each averaged over three technical replicates). Error bars are standard error of the mean. \*\*\*p<0.001; \*\*p<0.01; \*p<0.05 (ANOVA plus post hoc). (**f**) Transcriptional responses in different strains where *hap4* has been deleted, as measured by qPCR. Heat maps display mean values of log-fold changes observed across three biological replicates (each averaged over three technical replicates). UBC6 was used for normalization.
DOI: https://doi.org/10.7554/eLife.35330.013

p=5.5×10$^{-7}$), elevated oxygen consumption, and altered ATP and ROS production (*Figure 1*, *Figure 1—figure supplement 1*). In addition, we observe longer replicative life span (RLS) in $Mss116_{OE}$ (*Figure 1—figure supplement 4*), a more direct proxy of ageing that we were not able to measure accurately in $I_0$, where separating mother and daughter cells in a timely fashion proved challenging (see Materials and methods). Overexpression of a DEAD box mutant of Mss116 ($Mss116^{E268K}$), which lacks ATPase and therefore helicase activity, does not phenocopy $I_0$ (*Figure 1*). This suggests that the role of Mss116 in splicing – which relies on helicase activity – is critical rather than a recently suggested ATP-independent role in transcription elongation (*Markov et al., 2014*). More generally, the fact that $Mss116_{OE}$ – which encodes a full complement of introns – phenocopies $I_0$ indicates that the stress phenotype in these strains is not caused by missing DNA-level functionality and instead points toward a critical role for splicing. Overexpressing Mss116 in $I_0$ did not reveal additional phenotypes and the strain behaved like $I_0$ (*Figure 2—figure supplement 1*) further supporting the notion that intron deletion and Mss116 overexpression act through the same pathway.

## Cells are stressed because of abnormally efficient transcript maturation

We suspected that phenotypic effects of Mss116 overexpression (and intron deletion) are linked to altered transcript maturation of the host genes, *cob* and *cox1*. However, we first wanted to rule out an alternative hypothesis. In both $Mss116_{OE}$ and $I_0$, Mss116 is in excess relative to need (in $Mss116_{OE}$ because Mss116 is overexpressed, in $I_0$ because its usual targets – the introns – are absent). Phenotypic effects could therefore be caused by excess Mss116 interacting with RNAs that it would not normally target or not target to the same extent. This is conceptually related to the idea that splicing

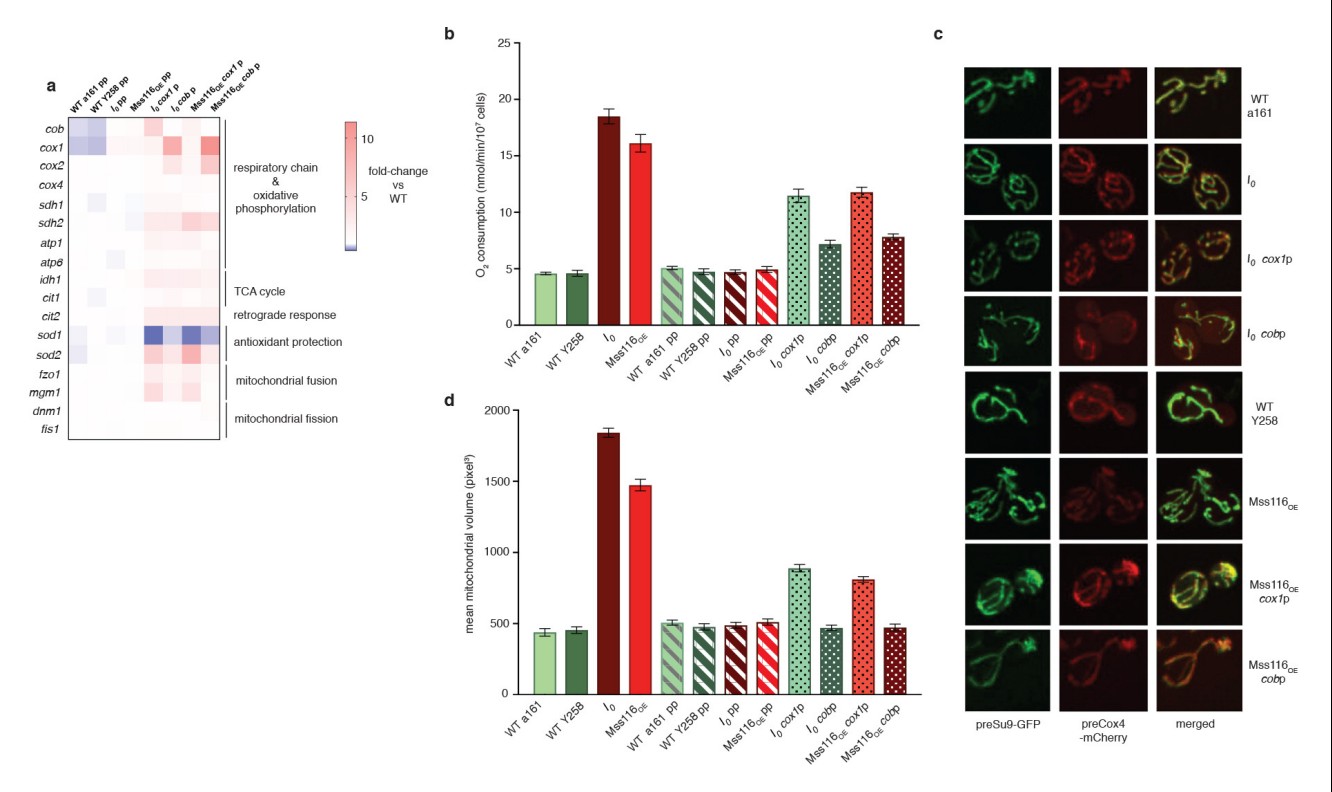

**Figure 5.** Dampening RNA levels of *cox1* and *cob* by reducing promoter activity partially rescues the intronless phenotype. (a) qPCR measurements in strains where either the *cob* promoter (cobp), the *cox1* promoter (cox1p) or both (pp) have been attenuated via targeted mutations. Heat maps display mean values of log-fold changes observed across three biological replicates (each averaged over three technical replicates). UBC6 was used for normalization. (b) Oxygen consumption and (c) mitochondrial morphology and (d) volume in response to promoter attenuation. Bar heights display the mean of three biological replicates (each averaged over three technical replicates). Error bars are standard error of the mean. Asterisks to indicate statistical significance are omitted for clarity. All comparisons between single- and double-promoter mutants and the corresponding parent strains are significant at p<0.001 (ANOVA plus post hoc).

DOI: https://doi.org/10.7554/eLife.35330.014

The following figure supplement is available for figure 5:

**Figure supplement 1.** Schematic representation of mutated positions in the promoter regions of *cox1* and *cob*.
DOI: https://doi.org/10.7554/eLife.35330.015

dynamics can change simply as a result of altered competition between mRNAs for access to the spliceosome (*Munding et al., 2013*). To test this hypothesis, we deleted Mss116 in the $I_0$ strain. We observe the same suite of phenotypes we see in $I_0$ (*Figure 2—figure supplement 2*), demonstrating that neither absolute nor relative excess of Mss116 are responsible for the stress phenotype described above.

Having ruled out this hypothesis, we then focused on characterizing the effects of Mss116 overexpression on *cob/cox1* splicing dynamics in greater detail. Using qPCR, we first measured the levels of total *cox1* and *cob* transcripts as well as individual introns at steady state (*Figure 2a*, 240 mins after Mss116 induction, see *Figure 2—figure supplement 3*). For *cox1*, we additionally monitored exon and intron (aI2, aI5β) levels using RNA fluorescent in situ hybridization (RNA FISH, *Figure 2b*, see Materials and methods). We find that most mitochondrial introns are strongly depleted in $Mss116_{OE}$ compared to the empty vector control and relative to exons. The relative depletion of individual introns is somewhat variable and one intron – aI5α - is equally abundant in $Mss116_{OE}$ and the Y258 WT (*Figure 2a*). We then examined shifts in the abundance of pre-mRNA and mature mRNA over the course of Mss116 induction, using different primer combinations to monitor unspliced RNAs and spliced exon-exon junctions. For both *cox1* and *cob*, Mss116 overexpression shifts the balance between unspliced (or partially spliced) pre-mRNA transcripts, which dominate the

uninduced steady state, towards mature mRNA transcripts (*Figure 2c*), while total transcript abundance (mRNA plus pre-mRNA) remains largely unchanged. These observations are consistent with a model where nascent transcriptional output is unchanged in $Mss116_{OE}$ compared to the empty vector control and differential steady state levels are the result of post-transcriptional events.

Based on these findings, we suggest that $Mss116_{OE}$ phenocopies $I_0$ because eliminating introns at the DNA level ($I_0$) and facilitating accurate and efficient excision at the RNA level ($Mss116_{OE}$) both result in abnormally efficient transcript maturation. That is, fewer transcripts are eliminated by mitochondrial quality control because splicing is erroneous or does not proceed in a timely manner, resulting in a greater number of mature *cox1/cob* mRNAs. For reasons that remain to be elucidated, increased transcript levels are then perceived as stressful and trigger the retrograde response, culminating in a multifaceted stress phenotype. We speculate in this regard that elevated protein levels of COB ($I_0$: 9.3-fold; $Mss116_{OE}$: 5.0-fold upregulation, as determined by quantitative label-free mass spectrometry) and COX1 ($I_0$: 11.6-fold; $Mss116_{OE}$: 12.1-fold upregulation) might interfere with proper assembly and function of complex III and complex IV, respectively, and therefore constitute a deleterious dosage imbalance phenotype.

## Discussion

It is now well documented that disruption of splicing homeostasis can impact normal physiological function and lead to cellular stress and disease (*Wang and Cooper, 2007*). There is also increasingly detailed mechanistic knowledge of how proteins involved in splicing can alter growth and ageing via a metabolic route, exemplified by the recent finding that splicing factor one is a modulator of dietary restriction-induced longevity in *Caenorhabditis elegans* (*Heintz et al., 2017*). The classic model here is that loss of splicing homeostasis – through genetic, developmental, or environmental perturbation – leads to deleterious shifts in splice isoform production or precipitates increased production of erroneous transcripts that tax the quality control system and/or have direct cytotoxic effects. In other words, the disease/stress state is a high-error state. Our results are unusual in that they suggest that normal splicing can be associated with high error rates and that, therefore, splicing homeostasis can also be disturbed by increasing splicing efficacy.

Further research will be required to tease apart how individual mitochondrial introns affect the overall burden from failed splicing in this system. It is evident from population genomic analysis of different *S. cerevisiae* strains that some mitochondrial introns are fixed across extant populations whereas others exhibit presence/absence polymorphism (*Wolters et al., 2015*). This seems to suggest that the removal of at least some introns in isolation is insufficiently stressful to be purged by natural selection. At the same time, studies of *suv3,* the second DEAD box RNA helicase present in yeast mitochondria, suggest that the deleterious effect of deleting individual introns – while possibly idiosyncratic – is at least partially cumulative: deletion of *suv3,* a component of the mitochondrial degradosome, decreases levels of mature *cox1/cob* mRNA and compromises respiratory capacity, but less so where more introns had been removed from the mitochondrial DNA (*Golik et al., 1995*). Importantly – at least for the combinations tested – severity was found to depend on the number but not identity of the *cox1* or *cob* introns present.

In addition to providing a new insight into post-transcriptional gene regulation in mitochondria, our findings have implications for understanding the evolutionary persistence of self-splicing introns and perhaps mobile elements more generally. The phenotypic effects we observe run counter to the notion that self-splicing introns are low-cost passengers and instead demonstrate that at least some of these selfish elements are firmly embedded in the organization of mitochondrial gene expression such that removing them upsets proper expression of their host genes. We suggest that our observations can be explained by an evolutionary lock-in model where the primordial colonization of an intron-free *cox1/cob* ancestor by a self-splicing intron led to a drop in *cox1/cob* mRNA levels and favoured compensatory mutations that increased *cox1/cob* transcription to restore mRNA abundances back to their original levels. When we forcibly remove these introns, however, this hard-wired upregulation turns maladaptive. There no longer is a pool of transcripts targeted for degradation leading to excess levels of mature mRNA. In principle, it is also possible that the initial invader spliced very efficiently and imposed no cost but subsequently co-evolved with host gene expression in a ratchet-like fashion, whereby incidental greater-than-required levels of the host gene allowed recurrent small decreases in splicing efficiency. However, since self-splicing introns can spread

despite substantial fitness costs (*Hickey, 1982*), we do not actually need to evoke a cost-free ancestral event. We suggest that evolutionary lock-ins of this type might provide an unappreciated mechanism to facilitate the longer term persistence of genetic parasites, especially in large host populations where evolution is not mutation-limited. We also note that this argument might in principle extend to nuclear introns: if, for a given dosage-sensitive gene, failure to splice is common and transcription levels are set to compensate, intron loss might be deleterious and prevented by purifying selection even though the intron makes no adaptive contribution to gene regulation.

## Materials and methods

### Strains and growth conditions

Strain a161 (also known as ID41-6/161, or sometimes simply 161), described by *Wenzlau et al. (1989)*, and the intronless a161-U7 ($I_0$) were gifts from Alan Lambowitz. These strains were used previously to show that splicing of group I and II introns is Mss116-dependent (*Huang et al., 2005*). a161 and a161-U7 are isogenic except for the mitochondrial genome and a single marker gene (a161: MATa ade1 lys1; a161-U7: MATa ade1 lys1 ura3).

Strain Y258 and the pBG1805 plasmid bearing *Mss116* for overexpression were purchased from Thermo Scientific (Dharmacon, Lafayette, CO). $Mss116^{E268K}$ was purchased from DNA 2.0 and cloned into pBG1805 using standard cloning techniques (*Sambrook and Maniatis, 1989*). Mss116 and $Mss116^{E268K}$ were overexpressed in the Y258 nuclear background. The expression of Mss116 and $Mss116^{E268K}$ was induced from the plasmids using 2% galactose (final), added at OD 0.2.

All strains were grown on YPD medium with 2% (w/v) glucose at 30°C with shaking. All experiments were performed on exponentially growing cells: cells were grown to OD 0.6–0.7 for WT, $I_0$, ΔHap4, and ΔRtg2 and to OD 0.9–1.0 for $Mss116_{OE}$ and $Mss116^{E268K}$, harvested by centrifugation at $4000 \times g$ for 5 min, washed and further treated as required.

In order to test growth on glycerol, strains were grown in YPD medium until saturation, at 30°C with shaking. Stationary cells were serially diluted and 5 µL drops plated onto YPEG agar plates (containing 3% ethanol and 3% glycerol). Growth was observed after 4 days.

### Gene deletion

Deletion of *hap4* and *rtg2* was performed as previously described (*Rothstein, 1983*), using a hygromycin cassette for selection in the WT background and a nourseothricine cassette in the $I_0$ background. Primers used for the deletions are listed in *Supplementary file 1*.

### Measurement of splicing kinetics in $Mss116_{OE}$

Overnight cultures of $Mss116_{OE}$ were diluted to OD 0.1. Galactose (2% final) was added at OD 0.2 to induce the expression of Mss116, and this is designated as time 0. Aliquots of the culture were harvested at 30, 60, 90, 120, 135, 150, 180, 210 and 240 min post-induction. The cells were pelleted and used for RNA isolation and cDNA preparation for qPCR. Multiple primer pairs were used to monitor unspliced intron-exon fragments and spliced exon-exon junctions (*Figure 2—figure supplement 4*, *Supplementary file 1*). Additional aliquots were harvested at 60, 150, 300, 360, 420, and 540 min post-induction and were used to measure Mss116 expression levels by flow cytometry (see below).

### Insertion of point mutations into the promoter regions

In order to introduce promoter-attenuating mutations into mitochondrial DNA, we followed the protocol described in *Bonnefoy et al., 2007*) for the integration of altered mtDNA sequences by homologous double crossovers. Briefly, a mutant fragment of mtDNA (in this case promoter sequences) flanked by WT mtDNA sequence is first transformed into a $rho^0$ strain, which is then mated with a recipient $rho^+$ strain. Upon mating, mitochondria from the two strains fuse and recombination between the two mtDNAs produces recombinant $rho^+$ strains in which the new mtDNA sequence is integrated by double crossover. For transformation, tungsten powder was used as a carrier of DNA (Tungsten M-10 Microcarriers #1652266, BioRad, Hercules, CA). Bombardment was performed using the Biolistic PDS-1000/He particle delivery system (BioRad). Cells were transformed with linear DNA fragments obtained by ligation of each mutated promoter region with 500 bp of up- and

downstream flanking DNA (*Supplementary file 1*). SacI and SalI restriction sites were added by PCR for ligation between the 3'-end of the upstream flanking region and the 5'-end of the promoter sequence, and 3'-end of the promoter sequence and 5'-end of the downstream flanking region, respectively. The mutations introduced here (highlighted in *Figure 5—figure supplement 1*) have been previously shown to reduce the strength of *cox1* and *cob* promoters (*Turk et al., 2013*).

## Chronological lifespan measurement

All strains were grown to saturation as described above and pelleted at 4000 $\times$ $g$ for 5 min. Cells were then washed twice and resuspended in sterile deionized water ($10^6$ cells in 10 mL in order to avoid cell growth on the debris of dead cells) and incubated at 30°C with shaking. Every 2–3 days, cells were serially diluted and plated onto YPD plates in order to evaluate cell growth.

## Replicative lifespan measurement

Replicative lifespan (RLS) was determined by micromanipulation for $Mss116_{OE}$ and $Mss116^{E268K}$ in the Y258 background, as well as for $Mss116_{OE}$ in the $\Delta hap4$ and $\Delta rtg2$ nuclear backgrounds, counting the number of daughters produced by individual mother cells. We were unable to reliably determine RLS for the $I_0$ strain as daughter and mother cells could not be separated in a timely manner, which is critical for RLS measurements. For unknown reasons, and perhaps specific to the nuclear background, cells were unusually sticky so that the first daughter often could not be separated from the mother until the mother had already produced other buds, making it difficult to track mother/daughter identity over time, an essential prerequisite for reliably determining replicative lifespan. Cells were incubated at 30°C on YPD (WT) or -URA (mutants) plates for the duration of the experiment. Using a microscope equipped with a microdissection apparatus suitable for *S. cerevisiae* (Singer Instruments, Watchet, UK), cells were transferred to defined places on agar plates and virgin daughter cells collected. Each cell was monitored continuously over several days every 60–90 min until all mother cells stopped budding. The total number of daughter cells was noted for each mother cell. The total number of monitored mother cells is as follows: 90 cells for the empty vector control, 86 cells for $Mss116_{OE}$, and 91 cells for the $Mss116^{E268K}$. The measurements were pooled from three independent experiments.

## Respiration measurement

Oxygen uptake was monitored polarographically using an oxygraph equipped with a Clark-type electrode (Oxygraph, Hansatech, Norfolk, UK). Cells were harvested during exponential growth phase, spun and resuspended in growth medium (as above) at the density of $30 \times 10^6$ cells/mL. 500 µL of culture were transferred to an airtight 1.5 mL oxygraph chamber. Cells were assayed in conditions closely similar to the ones in a flask culture (30°C and stirring). Oxygen content was monitored for at least 4 min. To ensure that the observed oxygen consumption was due to the mitochondrial activity, complex III inhibitor antimycin (final concentration 10 µg/mL) was routinely added to the cultures and compared to the rate observed without antimycin.

## Competition assay

Competition experiments were carried out between $I_0$ (a161-U7) and its control strain (a161) as well as between $Mss116_{OE}$ and its corresponding empty vector control strain. Prior to the competition, we plated $5 \times 10^9$ cells of each mutant ($I_0$ or $Mss116_{OE}$) strain on YPD agar plates with 200 µg/mL geneticin, followed by a 4-day incubation at 30°C, to select for spontaneous geneticin resistance. In doing so, we can subsequently determine relative fitness in a relatively simple fashion using a plating method (rather than, for example, sequencing barcodes). The two strains to be competed were then grown independently on YPD medium in 2% glucose until saturation. The next day, an equal number of cells from each WT culture (geneticin-sensitive) and each mutant (geneticin-resistant) were mixed in fresh YPD medium, 2% glucose so that each was diluted 200x. In mid-exponential phase, OD 0.4–0.6 (approximately 5–6 hr after dilution), aliquots of cells were harvested, and serial dilutions were plated on YPD-agar plates without geneticin. Next, dilutions with between 50 and 200 colonies were replica-plated on YPD-agar plates with geneticin (200 µg/mL). Colonies were counted on both types of plates and the ratio of geneticin-resistant (mutant) to geneticin-sensitive (WT) colonies was calculated as a measure of relative fitness.

## Flow cytometry

Flow cytometry was carried out on a Becton-Dickinson (Franklin Lakes, NJ) FACSCalibur machine equipped with a 488 nm Argon laser and a 635 nm red diode laser.

## Measurement of the Mss116 overexpression level

The expression level of Mss116 in the $Mss116_{OE}$ strain was measured by using a rabbit polyclonal anti-His tag antibody (Abcam, ab137839, 1:10000) and secondary IgG goat anti-rabbit labeled with Alexa 488 (Thermo Fisher Scientific, Waltham, MA, A11034, 1:2000). The signal obtained by flow cytometry (mean fluorescence over 10000 cells) was compared to the Mss116 tagged with GFP (Thermo Fisher Scientific) endogenous expression level estimated by using flow cytometry measurement based on the GFP signal. The mean fluorescence intensity in $Mss116_{OE}$ was normalized to the mean fluorescence intensity detected in wild-type cells with endogenous expression of Mss116.

## Mass spectrometry - sample processing

Isolated mitochondrial fractions containing 100 µg of protein were loaded onto Microcon 30kD centrifugal filters (Merck Millipore, Burlington, MA, MRCF0R030). Samples were then digested using a Filter Aided Sample Preparation (FASP) protocol (*Wiśniewski et al., 2009*). Briefly, samples were concentrated on the filter unit by centrifugation and buffer exchanged using sequential washing and centrifugation with 8M urea, 100 mM TRIS/HCL buffer (pH8.5). Proteins were reduced and alkylated sequentially with 10 mM Dithiothreitol and 50 mM Iodoacetamide (in 8M urea buffer), respectively. Samples were buffer exchanged to remove salts using sequential washing with 50 mM ammonium bicarbonate (AmBic). Trypsin Gold (Promega, Fitchburg, WI, V5280) was added to the samples in 50 mM ammonium bicarbonate to an approximate 1:50, protease:protein ratio. Digestions were incubated at 37°C overnight (17 hr). Digest extracts were recovered from FASP filters via centrifugation and acidified with 1% trifluoroacetic acid (TFA). Acidified protein digests were desalted using Glygen C18 spin tips (Glygen Corp, Columbia, MD, TT2C18.96) according to the manufacturer's recommendation and peptides eluted with 60% acetonitrile, 0.1% formic acid (FA). Eluents were then dried using a vacuum centrifuge.

## Mass spectrometry - liquid chromatography-tandem mass spectrometry (LC-MS/MS) analysis

Protein digests were redissolved in 0.1% TFA by shaking (1200 rpm) for 30 min and sonication on an ultrasonic water bath for 10 min, followed by centrifugation (14,000 rpm, 5°C) for 10 min. LC-MS/MS analysis was carried out in technical duplicates (1 µg on column) and separation was performed using an Ultimate 3000 RSLC nano liquid chromatography system (Thermo Fisher Scientific) coupled to a Orbitrap Velos mass spectrometer (Thermo Fisher Scientific) via an Easy-Spray source. For LC-MS/MS analysis protein digests were injected and loaded onto a trap column (Acclaim PepMap 100 C18, 100 µm × 2 cm) for desalting and concentration at 8 µL/min in 2% acetonitrile, 0.1% TFA. Peptides were then eluted on-line to an analytical column (Easy-Spray Pepmap RSLC C18, 75 µm × 50 cm) at a flow rate of 250 nL/min. Peptides were separated using a 120 min gradient, 4–25% of buffer B for 90 min followed by 25–45% buffer B for another 30 min (composition of buffer B – 80% acetonitrile, 0.1% FA) and subsequent column conditioning and equilibration. Eluted peptides were analysed by the mass spectrometer operating in positive polarity using a data-dependent acquisition mode. Ions for fragmentation were determined from an initial MS1 survey scan at 30,000 resolution, followed by CID (Collision-Induced Dissociation) of the top 10 most abundant ions. MS1 and MS2 scan AGC targets were set to 1e6 and 3e4 for maximum injection times of 500 ms and 100 ms respectively. A survey scan m/z range of 350–1500 was used, normalised collision energy set to 35%, charge state screening enabled with +1 charge states rejected and minimal fragmentation trigger signal threshold of 500 counts.

## Mass spectrometry - raw data processing

Data was processed using the MaxQuant software platform (v1.5.6.0), with database searches carried out by the in-built Andromeda search engine against the Uniprot *S. cerevisiae* database (version 20160815, number of entries: 6,729). A reverse decoy database approach was used at a 1% false discovery rate (FDR) for peptide spectrum matches. Search parameters included: maximum missed

cleavages set to 2, fixed modification of cysteine carbamidomethylation and variable modifications of methionine oxidation, asparagine deamidation and N-terminal glutamine to pyroglutamate conversion. Label-free quantification was enabled with an LFQ minimum ratio count of 2. 'Match between runs' function was used with match and alignment time limits of 1 and 20 min, respectively. Data have been deposited in the PRIDE repository [project accession number PXD008785].

## Assessment of mitochondrial membrane potential and mass

Variations of the mitochondrial transmembrane potential ($\Delta\Psi$m) were studied using 3,3-dihexyloxa-carbocyanine iodide [DiOC6(3)]. This cyanine cationic dye accumulates in the mitochondrial matrix as a function of $\Delta\Psi$m (*Perry et al., 2011*). Cells ($1 \times 10^6$/mL) were incubated in 1 mL culture medium containing 40 nM DiOC6(3) for 30 min in the dark at 30°C with constant shaking. DiOC6(3) membrane potential-related fluorescence was recorded using FL1 height. A total of 10,000 cells were analyzed for each curve. The collected data was analyzed using FlowJo software version 7.2.5 to determine the mean green fluorescence intensity after each treatment. The results are expressed as a percentage of mean fluorescence of the control strain. As a negative control, in each experiment, we preincubated aliquots of cells with carbonyl-cyanide 4-(trifluoromethoxy)- phenylhydrazone (FCCP, Sigma) and antimycin (Sigma, St. Louis, MO) at 100 µM and 5 µg/mL, respectively, 10 min before fluorescent dye staining, which leads to a collapse of mitochondrial membrane potential.

To measure mitochondrial mass, we used 10-N-Nonyl acridine orange (NAO), a dye that binds to cardiolipin, a phospholipid specifically present on the mitochondrial membrane (*Perry et al., 2011*). Cells ($1 \times 10^6$/mL) were incubated in 1 mL culture medium containing 100 nM NAO for 30 min in the dark at 30°C with constant shaking, followed by analysis on FACSCalibur flow cytometer with the same photomultiplier settings as used for DiOC6(3).

## Evaluation of the mitochondrial morphology and protein import machinery

To image mitochondrial morphology, strains were transformed with a MitoLoc plasmid (*Vowinckel et al., 2015*) (a gift from Markus Ralser) according to a previously described protocol (*Gietz and Schiestl, 2007*), with the only difference that the cells were incubated with the plasmid overnight at room temperature. Microscope slides were prepared as follows: 150 µL of YPD media containing 2% agarose was placed on a preheated microscope slide and cooled, before applying yeast cells to obtain a monolayer. The cells were centrifuged at 4000 × g for 3 min, and resuspended in 50 µL YPD. Once dry, the cover slip was placed, sealed, and mounted on a temperature-controlled Nikon Ti-E Eclipse inverted/UltraVIEW VoX (Perkin Elmer, Waltham, MA) spinning disc confocal setup, driven by Volocity software (version 6.3; Perkin Elmer). Images were recorded through a 60xCFI PlanApo VC oil objective (NA 1.4) using coherent solid state 488 nm and 543 nm diode lasers with a DPSS module, and a 1000 × 1000 pixel 14-bit Hamamatsu (C9100-50) electron-multiplied, charge-coupled device (EMCCD). The exposure time was 100 ms for GFP and 300 ms for mCherry, at 5–10% laser intensity. The number of cells with cytosolic mCherry accumulation was counted manually. More than 1000 cells were examined for each strain. Images were analyzed using ImageJ software with the MitoLoc plugin.

## ROS measurement

Cells were incubated in the dark with 5 µM MitoSOX red mitochondrial superoxide indicator (Molecular Probes, Eugene, OR) for 10 min at 30°C and subsequently analyzed by flow cytometry. Fluorescence (excitation/emission maxima of 510/580 nm) of 10,000 cells resulting from the intracellular red fluorescence was measured in the FL2 channel. The collected data was analyzed using FlowJo software version 7.2.5 for Microsoft (TreeStar, San Carlos, CA) to determine the mean green fluorescence intensity after each treatment. The results are expressed as the mean fluorescence across 10,000 cells.

## RNA extraction

Total RNA was isolated using the NucleoSpin RNA kit (Macherey and Nagel, Dueren, Germany) according to the manufacturer's instructions for up to $3 \times 10^8$ yeast cells, which includes incubation

with 50–100U of zymolyase for 1 hr at 30°C. The quality of the resulting total RNA was tested on 1% agarose gels.

## Genomic DNA isolation

Cells from saturated cultures (approximately $10^9$ cells) of WT Y258, WT a161, $I_0$, and $Mss116_{OE}$ were harvested by centrifugation and washed, as described above. Cell wall was digested for 1 hr at 30°C in the presence of 50–100U of zymolyase. Spheroplasts were then resuspended in 500 µL of cell lysis buffer (75 mM NaCl, 50 mM EDTA, 20 mM HEPES pH 7.8, 0.2% SDS). Next, 10 µL of 20 mg/mL proteinase K was added and the mixture incubated for 2 hr at 50°C. DNA was precipitated by the addition of isopropanol at room temperature, followed by centrifugation at 4°C and 11,000 $g$ for 30 min. The DNA pellet was then washed with ice-cold 70% ethanol, air-dried and resuspended in 50 µL of DNase-free water at 55°C.

## Quantitative real-time PCR (qPCR)

cDNA was synthesized from 1000 ng of total RNA using the iScript cDNA Synthesis Kit (Biorad). The cDNA was diluted 100-fold, mixed with primer pairs for each gene and SYBRgreen (BioRad). All primer pairs were designed to have a melting temperature of 60°C and are listed in *Supplementary file 1*. The qPCR reaction was run on a QuantFlexStudio 6 (Life Technologies, Carlsbad, CA) using 40 cycles, after which the melting curves for each well were determined. Final fold change values were estimated relative to the UBC6 gene in the control strain replicates. mtDNA copy number was assessed by qPCR using genomic DNA as template and primers against *cox1* and *cox3* (mtDNA) and *rpl32* (nuclear DNA, for normalization).

## Single-cell generation time measurement

Individual cells (approximately 100 for each strain) were placed on agar plates of appropriate growth medium, as described above, using a micromanipulator. Next, an image of each original mother cell was taken every 10 min for 8–9 hr. The images were then analyzed and division time of each cell was extracted.

## RNA-FISH and imaging

Yeast cultures were grown as described above, fixed with 37% formaldehyde for 45 min at room temperature, digested with 2.5 µL of zymolyase (Zymo Research, Irvine, CA, 2000 U) at 30°C for 60 min and permeabilized with 70% ethanol overnight at 4°C. Cells were hybridized in the dark at 30°C using Stellaris RNA-FISH probes (Biosearch Technologies, Novato, CA). Forty-five probes targeting intron aI2 and 40 probes targeting intron aI5b *cox1* were coupled to Quasar 670 dye (red). Forty-three probes targeting *cox1* exons were coupled to Quasar 570 dye (green). Yeast cells were placed on microscope slides with Vectashield Mounting Medium and imaged with an Olympus IX70 widefield fluorescence microscope. A series of z-stacks was acquired with a step size of 0.3 µm. The images were analysed using Image J. The number of green (exon), red (intron) and yellow (colocalized) foci was manually counted and normalized per 100 cells in each of three biological replicates. At least 300 cells were analyzed per replicate per strain.

## Acknowledgements

The authors would like to thank Tea Copic, Dirk Schwitters, Iva Pesun Medjimorec, Martin Billman, Sam Marguerat, and Xi Ming Sun for experimental assistance, Holger Kramer and the LMS Proteomics facility for proteomics work, and Nuno Raimundo, Ira Milosevic, Santiago Vernia, and Laurence Hurst for discussions. This work was supported by funding from the NAOS Group and the Mediterranean Institute for Life Sciences (MR, MM, MK, AK), an Imperial College Junior Research Fellowship (TW) and UK Medical Research Council core funding (TW).

## Additional information

### Funding

| Funder | Grant reference number | Author |
|---|---|---|
| NAOS Group | | Marina Rudan<br>Marina Musa<br>Matea Kanunnikau<br>Anita Kriško |
| Mediterrenean Institute of Life Sciences | | Marina Rudan<br>Marina Musa<br>Matea Kanunnikau<br>Anita Kriško |
| Imperial College London | Junior Research Fellowship | Tobias Warnecke |
| Medical Research Council | Core funding | Tobias Warnecke |

The funders had no role in study design, data collection and interpretation, or the decision to submit the work for publication.

### Author contributions

Marina Rudan, Conceptualization, Resources, Supervision, Funding acquisition, Investigation, Visualization, Writing—original draft, Writing—review and editing; Peter Bou Dib, Marina Musa, Matea Kanunnikau, Formal analysis, Investigation, Writing—review and editing; Sandra Sobočanec, Funding acquisition, Investigation; David Rueda, Conceptualization, Resources, Supervision, Funding acquisition, Investigation; Tobias Warnecke, Conceptualization, Resources, Formal analysis, Supervision, Funding acquisition, Visualization, Writing—original draft, Writing—review and editing; Anita Kriško, Conceptualization, Resources, Formal analysis, Supervision, Funding acquisition, Investigation, Visualization, Writing—original draft, Writing—review and editing

### Author ORCIDs

Peter Bou Dib (iD) http://orcid.org/0000-0002-7146-8271
Tobias Warnecke (iD) http://orcid.org/0000-0002-4936-5428
Anita Kriško (iD) https://orcid.org/0000-0001-7273-0190

### Decision letter and Author response

Decision letter https://doi.org/10.7554/eLife.35330.021
Author response https://doi.org/10.7554/eLife.35330.022

## Additional files

### Supplementary files

• Supplementary file 1. Compiled list of oligonucleotide sequences used in this study. The file consist of lists of qPCR and cloning primers, as well as sequences of RNA FISH probes, as separate tabs.
DOI: https://doi.org/10.7554/eLife.35330.016

• Transparent reporting form
DOI: https://doi.org/10.7554/eLife.35330.017

### Major datasets

The following dataset was generated:

| Author(s) | Year | Dataset title | Dataset URL | Database, license, and accessibility information |
|---|---|---|---|---|
| Tobias Warnecke | 2018 | Saccharomyces cerevisiae mitochondrial fractions by LC-MS/MS | https://www.ebi.ac.uk/pride/archive/projects/PXD008785 | Publicly available at EBI PRIDE (accession no. PXD008785) |

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
