## [Decision Letter]

Thank you for submitting your work entitled "Normal mitochondrial function in *Saccharomyces cerevisiae* requires inefficient splicing" for consideration by *eLife*. Your article has been reviewed by three peer reviewers, and the evaluation has been overseen by a Reviewing Editor, Timothy Nilsen, and a Senior Editor. The reviewers have opted to remain anonymous.

Our decision has been reached after consultation between the reviewers. Based on these discussions and the individual reviews below, we regret to inform you that your work in its current form cannot be considered further for publication in *eLife*. However, as you will see in the reviews below, all thought that the work was of potentially high interest. Therefore, we encourage you to address the reviewers' comments thoroughly and if successful to resubmit the paper to *eLife*. While such a resubmission will be considered to be a new submission, it will to the extent possible be handled by the same Editors and sent to the original reviewers.

Reviewer #1:

The authors show that deleting all self-splicing introns in the yeast mitochondria leads to a pronounced phenotype, and that this effect can be phenocopied by overexpressing Mss116. The work done appears to be solid although the writing and explanation of figures could be improved.

This reviewer's concern comes primarily from the assertion that the observed phenomena arise from increased expression of the intron-containing genes. Another possibility is that overexpression of Mss116 relative to 'need' leads to problems, for instance interference with expression of other genes. (This is related to Manny Ares' idea of 'hungry spliceosomes' wherein overexpression of spliceosomes relative to number of intron-containing transcripts can lead to aberrant splicing of other (non-intron-containing) transcripts; it is also related to the notion that having extra proteins around is costly, as Drummond et al., have been arguing, although there is a bit of a difference here as that work concerns proteins without functions). Fortunately, this should be easy to test - does partial or complete KD of Mss116 ameliorate the phenotypes?

In addition, I think that the authors somewhat simplify the possible pathways to what they observe, thus setting up a bit of a strawman. Given that stuff has to be spliced, it is actually not all that surprising that splicing has become integrated. This is an important lesson, though one not as surprising as the paper perhaps tries to let on; it is not surprising that there will be coevolution of machinery and substrate such that removal of a network component would screw things up. As such I don't regard this work as a particularly 'clean' a test of repetitive elements' selective effects.

In particular, in the Results and Discussion section. The suggested model is possible but postulates a costly mutation (one that substantially decreases host gene expression). Perhaps instead an efficiently spliced intron invaded and gradually lost efficiency little by little while it was compensated by increased gene expression. If a non-infinitesimal band of gene expression is well tolerated, we expect the transcriptional level to float within this range (more precisely, within the range of transcriptional activity that, coupled with the splicing efficiency at the time). If there is mutational pressure imposes a direction on likely evolution of splicing efficiency (that is, if efficiency-decreasing mutations are more likely than -increasing mutations, which seems quite likely), then transcriptional level could float to a position of (non-selected) excess expression, which could allow fixation of a splicing efficiency decreasing mutation, and multiple cycles of this could lead to increased transcription and inefficient splicing without inferring a costly intermediate).

In total, then, I think that the authors should produce a draft of the manuscript that is more mindful of the limitations of their experiment, notwithstanding its elegance, to test the idea that (relative) overexpression of Mss116 is the problem rather than overexpression of intron-containing transcripts, and to spend more time discussing how surprising their findings really are given what we already know. After this revision I recommend that the manuscript be fully reconsidered; while I am hopeful that the paper will eventually reach a level of appropriate consideration of alternative possibilities and models, I hope that his partial encouragement will not be misunderstood as full endorsement and look forward to reviewing an improved and more subtle draft of the manuscript.

My hopes for improvement notwithstanding, I congratulate the authors on some very nice work.

Reviewer #2:

Summary

The authors have conducted an interesting study that suggests that deleting the complete repertoire of group I and group II introns from the yeast mitochondrial genome leads to phenotypes consistent with a retrograde response. Intriguingly, a similar phenotype also is observed upon over-expression of the Mss116 helicase, but not upon over-expression of a mutant Mss116 protein that lacks ATPase (and therefore helicase) activity. These (and other) results lead the authors to conclude that normal mitochondrial function "depends" on inefficient splicing and that either the absence of introns or over-expression of Mss116 leads to abnormally efficient transcript maturation, thereby increasing the levels of mature cob and cox1 RNA, which accounts for the phenotypes observed in their studies. The authors then propose a rather intriguing evolutionary "lock-in" model. The model posits that the colonization of the primordial cob and coxI genes by mobile group I and/or group II introns led to inefficient splicing which, in turn, led to compensatory regulatory mechanisms that increased cob and coxI expression to allow efficient oxidative phosphorylation. The forceful experimental removal of the mitochondrial introns (or promoting more efficient splicing by over-expression of Mss116) then is predicted to lead to an over-expression of cob and coxI, which the authors demonstrate compromises mitochondrial function.

For the most part, this paper is well written, the available data support the conclusions, and the resultant model is intriguing. That being stated, additional editing and the inclusion of important controls would help clarify the presentation and strengthen the conclusions of the study. My suggestions for improvement are noted below.

Essential revisions:

1) Title: Does mitochondrial function truly "require" inefficient splicing? It seems more likely that the population of the primordial cob and coxI genes by mobile introns was slightly detrimental, which allowed compensatory mutations that resulted in higher cob and coxI expression. If so, these data are consistent with the idea that mobile intron insertions were detrimental and that the host evolved second site mutations to restore cob and coxI expression. Thus, the artificial, forced deletion of the introns then leads to an over-accumulation of cob and coxI (which is the crux of the "lock-in" model). Though semantic, I have a hard time buying that the presence of introns and inefficient splicing are required for mitochondrial function. Instead, one can simply argue that mobile introns are successful parasites that disrupted the cob and coxI genes and that the host "found" a way to overcome the burden of intron insertion. Long story short-I have issues with the words "requires" or (in the Abstract) "is dependent" on inefficient splicing.

2) Results and Discussion section: Does the intron-minus strain, when compared to the wt strain, exhibit a growth defect on non-fermentable carbon sources?

3) Why is oxygen consumption increased in the intron-minus strain, whereas growth is decreased?

4) Results and Discussion section: idh1 expression is not shown in Figure 2A.

5) In general, the figures do not strictly follow the text, which caused me to jump between figures while reading the paper. Can the authors improve the presentation in this regard?

6) Can the authors conduct simple chemostat-type experiments to clearly demonstrate that wt yeast can "out compete" intron-minus and Mss116 over-expression mutant yeast for nutrients in culture-such an experiment would, in my opinion, be more convincing than a growth rate experiment when considering the proposed evolutionary model.

7) Have the authors tested whether the over-expression of cob and coxI RNA lead to higher levels of cob and coxI proteins? For example, an S-35 labeling experiment in the presence of cycloheximide could be very informative in testing whether RNA over-expression leads to protein over-expression. Basically, is it over-expression of cob and coxI RNA and/or over-expression of cob and coxI protein that lead to the observed phenotypes-these points are not clear from my reading of the paper.

8) Though I appreciate the RNA FISH experiments shown throughout the paper, old-fashioned Northern blots would provide an orthologous means to look at intron accumulation cob and coxI RNA expression, etc. Can the authors consider conducting such experiments to examine transcript/intron accumulation?

9) Figure 1: The legend and Figure panels do not agree-please fix.

Reviewer #3:

This manuscript addresses the issue of whether group I and group II introns in yeast mitochondrial genomes have a physiological consequence, or whether they are essentially silent. The latter has been generally assumed, because many organellar introns are "optional", and no phenotypic differences have been observed among strains that contain or lack the optional introns. Still, it is tempting to speculate that the organellar introns have consequences that have gone unnoticed.

The authors start with a yeast strain in which all organellar introns have been deleted. They find that the strain grows more slowly than the wild type and has a number of physiological perturbations including an increase in mitochondrial mass in the cell, increased oxygen consumption, and activation of the retrograde response. Transcript levels of the cox1 and cob1 mitochondrial genes are elevated. To test if higher transcript levels per se are responsible for the phenotype, they make point mutations in the promoters of the two genes and find that the wild type phenotype is largely restored. Hypothesizing that the presence of the introns causes inefficient transcript maturation, they overexpress the helicase Mss116 in an intron-containing strain. Mss116 is known to facilitate splicing of organellar introns, and interestingly its overexpression causes a similar phenotype as removal of the introns. This is convincing evidence that the efficiency of intron splicing is responsible for the mitochondrial phenotype, and that the cell is adapted to accommodate low efficiency splicing, and that having higher efficiency transcript maturation causes stress and irregularities in the cells. Thus, the organellar introns have at least a biological consequence, if not an adaptive role.

Overall, the experiments and data are interesting and the conclusions significant. However, there are some issues with the presentation and possibly the data, because not all experimental details are clear to me. The main text is highly condensed and does not clearly explain the experiments or data. Conclusions are stated but not explained thoroughly. It is especially difficult to follow if one does not have a background in mitochondrial physiology and the associated assays. I expect that all the assays and data support the conclusions that are drawn, but I am not sure because the explanations are so brief or even left out.

The most important thing missing from the manuscript is a description and explanation of the intron-less strain. There is no mention of where the strain came from. Did the authors make it themselves? If so, how? How did they validate the strain? The issue of isogenicity of the intronless strain and the wild-type strain is not addressed. This is a very important issue of course because the entire manuscript is based on the strain.

There are also issues with the figures and legends. For example, in Figure 1 what are all the circles in panel A? Not all points in panel B have error bars. Not all of the descriptions match-up between the panels and legend. More care should be given to the presentation, for both the images and legends.

The authors seem to think that the presence of introns leads to mis-splicing of transcript, but isn't it more likely that maturation is simply slower because of the time it takes to splice? Is there any evidence for mis-splicing and degradation as opposed to just slow splicing?

---

## [Author Response]

Reviewer #1:The authors show that deleting all self-splicing introns in the yeast mitochondria leads to a pronounced phenotype, and that this effect can be phenocopied by overexpressing Mss116. The work done appears to be solid although the writing and explanation of figures could be improved.This reviewer's concern comes primarily from the assertion that the observed phenomena arise from increased expression of the intron-containing genes. Another possibility is that overexpression of Mss116 relative to 'need' leads to problems, for instance interference with expression of other genes. (This is related to Manny Ares' idea of 'hungry spliceosomes' wherein overexpression of spliceosomes relative to number of intron-containing transcripts can lead to aberrant splicing of other (non-intron-containing) transcripts; it is also related to the notion that having extra proteins around is costly, as Drummond et al., have been arguing, although there is a bit of a difference here as that work concerns proteins without functions). Fortunately, this should be easy to test -- does partial or complete KD of Mss116 ameliorate the phenotypes?

The reviewer proposes a reasonable alternative hypothesis: in both *I_0_* and *Mss116_OE_*, Mss116 is in excess relative to need (in *Mss116_OE_* because Mss116 is overexpressed, in *I_0_* because its usual targets – the introns – are absent). Might the phenotypic effects we observe therefore be explained by Mss116 starting to interact (or interacting more strongly) with off-target RNAs? We tested this experimentally. As suggested by the reviewer, we deleted Mss116 from the *I_0_* strain. In this *I_0_* /Δ*Mss116* strain, any phenotypic effects cannot possibly come from excess Mss116. Reassuringly, the phenotype of *I_0_* /Δ*Mss116* mirrors that of *I_0_*, rejecting this hypothesis. Note also that the “hungry Mss116” hypothesis is not consistent with the stress phenotype being lost when rendering the *cob/cox1* promoters hypoactive. The “hungry Mss116” hypothesis would have predicted no change or perhaps further exacerbation of the phenotype. We discuss these new results in the Results and Discussion section and include them as Supplementary Figure 6.

If we understand correctly, the reviewer is concerned about the costs of gratuitous expression, as exemplified by the work of Kafri et al., 2015 andStoebel et al., 2008 (Alan Drummond’s past work is more specifically concerned with the cost of misfolded proteins). The rationale here would be that expressing a protein – even if functionally inert – can be costly because it sequesters away cellular resources, e.g. by hogging ribosomes. We do not think that this is a concern here. First, any such generic cost of expression will be born both by the *Mss116_OE_* strain and the catalytic mutant (*Mss116^E268K^*), for which we do not observe the same suit of phenotypes. In other words, our study already contained an internal control. Second, the lack of phenotype in *I_0_* /Δ*Mss116* (see response to point #1 above) also rules out that a gratuitous expression cost of *Mss116_OE_*is responsible for the phenotype.

In addition, I think that the authors somewhat simplify the possible pathways to what they observe, thus setting up a bit of a strawman. Given that stuff has to be spliced, it is actually not all that surprising that splicing has become integrated. This is an important lesson, though one not as surprising as the paper perhaps tries to let on; it is not surprising that there will be coevolution of machinery and substrate such that removal of a network component would screw things up. As such I don't regard this work as a particularly 'clean' a test of repetitive elements' selective effects.

We agree that it might well be intuitive and expected (at least to some people) that splicing – even inefficient splicing – becomes integrated into the gene expression process so that intron removal interferes with proper expression. But intuition is one thing, demonstrating it is another. To the best of our knowledge, there is no prior demonstration of this phenomenon (certainly in such a well-defined context), that not only describes detailed physiological/fitness costs but also traces the molecular origins of these costs.

In particular, in the Results and Discussion section. The suggested model is possible but postulates a costly mutation (one that substantially decreases host gene expression). Perhaps instead an efficiently spliced intron invaded and gradually lost efficiency little by little while it was compensated by increased gene expression. If a non-infinitesimal band of gene expression is well tolerated, we expect the transcriptional level to float within this range (more precisely, within the range of transcriptional activity that, coupled with the splicing efficiency at the time). If there is mutational pressure imposes a direction on likely evolution of splicing efficiency (that is, if efficiency-decreasing mutations are more likely than -increasing mutations, which seems quite likely), then transcriptional level could float to a position of (non-selected) excess expression, which could allow fixation of a splicing efficiency decreasing mutation, and multiple cycles of this could lead to increased transcription and inefficient splicing without inferring a costly intermediate).

We agree that, in principle, the initial invader might have spliced efficiently and subsequently become less efficient, co-evolving with host gene expression in a ratchet-like fashion, whereby incidental increases in the expression level of the host gene allowed small decreases in splicing efficiency of the intron. We now acknowledge this scenario (Materials and methods section). However, since elements that operate like self-splicing introns can spread in the population despite substantial fitness costs (see article by Hickey mentioned by reviewer #3), it is arguably more parsimonious to not require perfect splicing upon initial invasion, a point we also highlight in the revised manuscript.

In total, then, I think that the authors should produce a draft of the manuscript that is more mindful of the limitations of their experiment, notwithstanding its elegance, to test the idea that (relative) overexpression of Mss116 is the problem rather than overexpression of intron-containing transcripts, and to spend more time discussing how surprising their findings really are given what we already know. After this revision I recommend that the manuscript be fully reconsidered; while I am hopeful that the paper will eventually reach a level of appropriate consideration of alternative possibilities and models, I hope that his partial encouragement will not be misunderstood as full endorsement and look forward to reviewing an improved and more subtle draft of the manuscript.

We apologize for the poor legend to Figure 1, which reflected an earlier version of the figure and therefore was incomplete and had a number of incorrect panel assignments, explaining why negative results appeared to be stated as positive results etc. We have revised all figure legends and taken extra care to explain individual elements, clarify the colour scheme and assign panels correctly.

Regarding hyperfusion, we argue that elevated rates of fusion (rather than impaired fission) explains the larger branched network morphology because regulators of mitochondrial fusion (*fzo1, mgm1*) are strongly upregulated, whereas genes controlling mitochondrial fission are comparatively static. We have clarified this outside the figure legend (Results and Discussion section).

My hopes for improvement notwithstanding, I congratulate the authors on some very nice work.Reviewer #2:Essential revisions:1) Title: Does mitochondrial function truly "require" inefficient splicing? It seems more likely that the population of the primordial cob and coxI genes by mobile introns was slightly detrimental, which allowed compensatory mutations that resulted in higher cob and coxI expression. If so, these data are consistent with the idea that mobile intron insertions were detrimental and that the host evolved second site mutations to restore cob and coxI expression. Thus, the artificial, forced deletion of the introns then leads to an over-accumulation of cob and coxI (which is the crux of the "lock-in" model). Though semantic, I have a hard time buying that the presence of introns and inefficient splicing are required for mitochondrial function. Instead, one can simply argue that mobile introns are successful parasites that disrupted the cob and coxI genes and that the host "found" a way to overcome the burden of intron insertion. Long story short-I have issues with the words "requires" or (in the Abstract) "is dependent" on inefficient splicing.

We are in perfect agreement with the reviewer about the likely evolutionary chain of events: One (or several) introns invade *cox1* and *cob*, compromising levels of mature mRNA. This sets the scene for compensatory mutations to arise that upregulate *cox1/cob* expression, allowing the host to overcome the burden of intron insertion. As an incidental consequence, introns become harder to lose because, in the absence of these introns, increased mRNA levels become detrimentally high.

Clearly, if we designed the system from scratch, introns would not be required for normal mitochondrial function. Thus, when we say “require” we refer not to an absolute requirement, but the current, evolved situation: the fact that inefficient splicing, despite its obvious wastefulness, has become integrated in the process of mitochondrial gene expression.

We understand the referee’s concern that using “require” (and “is dependent”) might evoke an absolute requirement rather than an evolved state and we have therefore rephrased this to “has become dependent on” (or similar) throughout the manuscript. We hope that this phrasing, although less punchy, better captures the notion of evolutionary lock-in, but we are open to suggestions on how to convey the phenomenon more eloquently.

2) Results and Discussion section: Does the intron-minus strain, when compared to the wt strain, exhibit a growth defect on non-fermentable carbon sources?

As we briefly alluded to in the previous version of the manuscript, prior work had already considered how *I_0_* grows on glycerol, a commonly used non-fermentable carbon source (Minczuk et al., 2002; Huang et al., 2005). Minczuk et al., reported easily visible but smaller colonies (for a different nuclear background) using a standard drop test, while Huang et al. did not comment on qualitative differences, simply stating that they found *I_0_* to be Gly+. We carried out our own drop tests for *I_0_*, *Mss116_OE_, Mss116^E268K^*, and the two control strains on glycerol- and glucose-supplemented medium. All strains were Gly+ and we observed no obvious qualitative differences, in line with the results of Huang et al. The results are consistent with our other phenotypic observations: intron removal (or more efficient splicing) does not terminally compromise mitochondrial function but rather triggers a compensatory stress response, which works pretty well, although quantitative assays do reveal growth and fitness deficits. We now include the results of our drop tests (Supplementary figure 2) and highlight how they support the finding that mitochondria remain functional (Results and Discussion section).

3) Why is oxygen consumption increased in the intron-minus strain, whereas growth is decreased?

Oxygen consumption is indicative of metabolic activity but not all of that activity needs to be converted into growth. Increased biogenesis of mitochondria and molecular components involved in the stress response might well explain elevated oxygen consumption in the absence of a concomitant increase in growth.

4) Results and Discussion section: idh1 expression is not shown in Figure 2A.

Idh1 expression is shown in Figure 2A right above cit1 (TCA cycle category).

5) In general, the figures do not strictly follow the text, which caused me to jump between figures while reading the paper. Can the authors improve the presentation in this regard?

We are aware that this is not ideal but have struggled to come up with a better solution. The most important consideration for us was having all strains presented alongside each other to allow direct comparisons, especially between *I_0_*and *Mss116_OE_*. We can only do this by either including all the information in one figure (as we currently do) or duplicating information (presenting *I_0_* first and then including the same *I_0_*information in a new *Mss116_OE_* figure), which seems redundant and therefore even less desirable. We hope that this is less of a nuisance once the manuscript has undergone conversion into a more compact format (or can be read online with clickable links to the relevant figures).

6) Can the authors conduct simple chemostat-type experiments to clearly demonstrate that wt yeast can "out compete" intron-minus and Mss116 over-expression mutant yeast for nutrients in culture-such an experiment would, in my opinion, be more convincing than a growth rate experiment when considering the proposed evolutionary model.

Even though growth rate measurements often serve as a reasonable proxy for fitness, we agree with reviewer that, given the evolutionary framing, a more direct read-out of differential fitness is desirable. We therefore carried out competition experiments between *I_0_* (a161-U7) and its control strain (a161) as well as between Mss116_OE_ and its corresponding empty vector control strain. Prior to the competition, we plated 5x10^Hickey, 1982^ cells of each mutant (*I_0_* or *Mss116_OE_*) strain on YPD agar plates with 200 μg/mL geneticin, followed by a 4-day incubation at 30^o^C, to select for spontaneous geneticin resistance. In doing so, we can subsequently determine relative fitness in a relatively simple fashion using a plating method (rather than, for example, competing in a chemostat and then sequencing barcodes). The two strains to be competed were then grown independently on YPD medium in 2% glucose until saturation. The next day, an equal number of cells from each WT culture (geneticin-sensitive) and each mutant (geneticin-resistant) were mixed in fresh YPD medium, 2% glucose so that each was diluted 200x. In mid-exponential phase, OD 0.4-0.6, aliquots of cells were harvested, and serial dilutions were plated on YPD-agar plates without geneticin. Next, dilutions with between 50 and 200 colonies were replica plated on YPD-agar plates with kanamycin (50 ug/mL). Colonies were counted on both types of plates and the ratio of geneticin-resistant (mutant) to geneticin-sensitive (WT) colonies was calculated as a measure of relative fitness. We include these results as Supplementary figure 1 and mention them in the Results and Discussion section.

7) Have the authors tested whether the over-expression of cob and coxI RNA lead to higher levels of cob and coxI proteins? For example, an S-35 labeling experiment in the presence of cycloheximide could be very informative in testing whether RNA over-expression leads to protein over-expression. Basically, is it over-expression of cob and coxI RNA and/or over-expression of cob and coxI protein that lead to the observed phenotypes-these points are not clear from my reading of the paper.

We have determined changes in COB and COX1 protein levels using quantitative label-free mass spectronomy. Both proteins are upregulated in *I_0_* (COX1 11.6-fold, COB 9.3-fold) and *Mss116_OE_* (COX1 12.1-fold, COB 5.0-fold), which is consistent with the idea that the stress phenotype is mediated by excess dosage at the protein level. We include these results in the Materials and methods section. However, we think it is important to stress that elevated protein levels do not directly implicate a protein-level over an RNA-level mechanism. To do so, one would have to specifically reduce *cox1/cob* RNA levels while leaving protein levels unchanged or vice versa. This is currently outside of our capabilities and the reason why – although it is tempting to suggest a protein-centric mechanism – we have confined ourselves to careful speculation in this regard and principally focused our attention on the ultimate cause of the phenotype: altered splicing.

8) Though I appreciate the RNA FISH experiments shown throughout the paper, old-fashioned Northern blots would provide an orthologous means to look at intron accumulation cob and coxI RNA expression, etc. Can the authors consider conducting such experiments to examine transcript/intron accumulation?

We agree with the reviewer that Northern blots would have provided an orthogonal means to understand expression/splicing dynamics. However, we already provide two orthogonal techniques (PCR and RNA FISH). Rather than adding a third, we have instead focused our efforts on carrying out experiments (requested above and below) that can provide new insights rather than additional confirmation.

9) Figure 1: The legend and Figure panels do not agree-please fix.

We have now fixed this issue. Please also see the response to comment #6 above.

Reviewer #3:The most important thing missing from the manuscript is a description and explanation of the intron-less strain. There is no mention of where the strain came from. Did the authors make it themselves? If so, how? How did they validate the strain? The issue of isogenicity of the intronless strain and the wild-type strain is not addressed. This is a very important issue of course because the entire manuscript is based on the strain.

We appreciate that we should have provided more information on the origin of the intronless strain and issues of isogenicity. We now do so (subsection “*Removal of mitochondrial introns is associated with a multi-faceted stress phenotype”* and subsection “Strains and growth conditions”).

Intronless mitochondria were originally constructed in Gérard Faye’s lab. By generating a series of mitochondrial recombinants between natural yeast isolates that lack individual mitochondrial introns, the authors managed to create a mitochondrial genome completely devoid of introns (Séraphin et al., 1987). This intronless mitochondrial genome has since been transduced into a number of different strains, including a strain known as *a161* (or *ID41-6/161*, or sometimes simply *161*), as described by (Wenzlau et al., 1989). This strain was prominently used by Alan Lambowitz to show that splicing of group I and II introns is Mss116-dependent (Huang et al., 2005). We decided to use this strain so that we could compare our results (regarding growth defects etc.) directly with prior literature. To this end, Alan Lambowitz kindly gifted us two strains: the *a161* strain with the intronless mitochondrial genome (sometimes referred to as *a161-U7*) and the *a161* control strain, whose mitochondrial genome contains the seven *cox1* and five *cob* introns that are also present in the Y258 strain used for Mss116 overexpression (see below). Strains *a161* (WT) and *a161-U7 (I_0_*) are isogenic except for the mitochondrial genome and a single marker gene (a161: MATa ade1 lys1; *I_0_*: MATa ade1 lys1 ura3).

Mss116 overexpression, on the other hand, was carried out in the Y258 nuclear background, commonly used for overexpression experiments. The three strains we used (WT control, *Mss116_OE_, Mss116^E268K^*) are isogenic except for the plasmid, which is either empty, contains Mss116, or the Mss116 ATPase mutant.

The reason why we ended up using two different nuclear backgrounds is two-fold: first, as already mentioned, we thought it prudent to use the *a161* strain to assess the properties of *I_0_* so we could usefully compare and sanity-check our results against previous work. The second reason is historical. We had previously done some work on Mss116 and already constructed the appropriate strains in the Y258 background.

The most important thing to highlight here is that each “intervention” (intron deletion or overexpression) has its own, appropriate, isogenic control.

NB: as evident from the results presented, the two control strains (a161 and Y258) behave very similarly in all phenotypic aspects we considered, but genetic differences might explain, for example, our difficulties in separating mother from daughter cells in the a161 background.

There are also issues with the figures and legends. For example, in Figure 1 what are all the circles in panel A? Not all points in panel B have error bars. Not all of the descriptions match-up between the panels and legend. More care should be given to the presentation, for both the images and legends.

Please see response to comment #6 above.

The authors seem to think that the presence of introns leads to mis-splicing of transcript, but isn't it more likely that maturation is simply slower because of the time it takes to splice? Is there any evidence for mis-splicing and degradation as opposed to just slow splicing?

We now present a time course of pre-mRNA/mRNA abundance following Mss116 induction that might help to clarify some of these issues (subsection “Cells are stressed because of abnormally efficient transcript maturation”). The time course shows that Mss116 induction dramatically shifts the balance between pre-mRNA and mature mRNA. Prior to induction, mature mRNA levels are low compared to pre-mRNA, implying that only a fraction of pre-mRNAs reaches maturity. Although we agree that Mss116 might affect maturation speed, it is important to highlight that, at steady state, speed per se would not explain differential mRNA abundances – there has to be degradation. How, then, does the presence of introns lead to elevated rates of degradation? Altered kinetics might indeed play a role, i.e. splicing might fail to occur in a timely manner so that the transcript then becomes a target for the mitochondrial degradosome. Alternatively, introns might never be spliced or get spliced imprecisely. When we talked about “mis-splicing” in the manuscript, this should be read to include all the above options (kinetic, failure to splice, incorrect splicing) that can call the degradation machinery into action. We have clarified this in subsection “The effects of intron removal are phenocopied by overexpression of Mss116” and also replaced the word “mis-splicing” with the broader term “failure to splice” throughout the manuscript.